

# Pollen metabarcoding reveals broad and species-specific resource use by urban bees

Caitlin Potter[1,2], Natasha de Vere[1,3], Laura E. Jones[3,4], Col R. Ford[3], Matthew J. Hegarty[1], Kathy H. Hodder[2], Anita Diaz[2] and Elizabeth L. Franklin[2,5]

[1] IBERS, Aberystwyth University, Aberystwyth, Ceredigion, UK
[2] Department of Life and Environmental Sciences, Bournemouth University, Poole, UK
[3] National Botanic Garden of Wales, Llanarthne, Carmarthenshire, UK
[4] Molecular Ecology and Fisheries Genetics Laboratory, Bangor University, Bangor, Gwynedd, UK
[5] School of Environmental Sciences, University of Guelph, Guelph, ON, Canada

## ABSTRACT

Bee populations are currently undergoing severe global declines driven by the interactive effects of a number of factors. Ongoing urbanisation has the potential to exacerbate bee declines, unless steps are taken to ensure appropriate floral resources are available. Sown wildflower strips are one way in which floral resources can be provided to urban bees. However, the use of these strips by pollinators in urban environments remains little studied. Here, we employ pollen metabarcoding of the *rbcL* gene to compare the foraging patterns of different bee species observed using urban sown wildflower strips in July 2016, with a goal of identifying which plant species are most important for bees. We also demonstrate the use of a non-destructive method of pollen collection. Bees were found to forage on a wide variety of plant genera and families, including a diverse range of plants from outside the wildflower plots, suggesting that foragers visiting sown wildflower strips also utilize other urban habitats. Particular plants within the wildflower strips dominated metabarcoding data, particularly *Papaver rhoeas* and *Phacelia tanacetifolia*. Overall, we demonstrate that pollinators observed in sown wildflower strips use certain sown foodplants as part of a larger urban matrix.

# INTRODUCTION

Over the last century, wild bee populations have suffered widespread declines in the form of substantial range contractions (*Kerr et al., 2015*) alongside local decreases in the abundance and species richness of hoverfly and bee communities (*Biesmeijer et al., 2006*; *Potts et al., 2010*; *Bommarco et al., 2011*; *Cameron et al., 2011*; *Dupont, Damgaard & Simonsen, 2011*; *Bartomeus et al., 2013*). Wild bee declines likely result from the interactive effects of multiple factors (*Goulson et al., 2015*), including habitat loss and fragmentation (e.g. due to urbanisation; *Garibaldi et al., 2011*; *Gerard et al., 2010*; *Hendrickx et al., 2007*; *NEA, 2011*), climate change (*Kerr et al., 2015*) and parasite and pathogen spread (*Otterstatter & Thomson, 2008*). Bee declines are of economic concern due to the value of

Corresponding author
Caitlin Potter, cap34@aber.ac.uk

pollination services to agriculture (*Klein et al., 2007*; *NEA, 2011*). While the global number of managed honeybee colonies has increased (*Aizen & Harder, 2009*), this is unlikely to be sufficient to compensate for wild pollinator losses: the rate of increase currently does not match the rate of increase in the need for pollinator services (*Aizen & Harder, 2009*) and certain regions are experiencing a reduction in numbers of beekeepers and managed honeybee colonies (*National Research Council, 2007*; *Potts et al., 2010*).

Urbanisation is increasing both within the UK and globally, with increasing housing density and population (*Gerard et al., 2010*; *NEA, 2011*; *Seto et al., 2011*). Continuing urbanisation, with associated displacement of semi-natural and agricultural habitats (*Gerard et al., 2010*; *NEA, 2011*), will potentially have further negative effects on bee populations. Compared to rural environments, urban areas may have lower bee species richness (*Hernandez, Frankie & Thorp, 2009*; *Bates et al., 2011*; *Deguines et al., 2012*), fewer plant–pollinator interactions (*Geslin et al., 2013*), and a lower abundance of pollinators (*Bates et al., 2011*). Conversely, other studies suggest neutral or even positive effects of urbanisation on pollinator species richness (*Banaszak-Cibicka & Zmihorski, 2012*; *Baldock et al., 2015*). The degree to which urban areas are able to support rich and abundant pollinator communities is related to the ability of these areas to provide the resources required to support wild bees and hoverflies, particularly floral resources (nectar and pollen) (*McFrederick & LeBuhn 2006*; *Bates et al., 2011*; *Fortel et al., 2014*; *Hülsmann et al., 2015*).

One method commonly advocated to enhance urban habitats for pollinators is the provision of sown wildflower plots. These plots provide significantly greater nectar resources than amenity grasslands (*Hicks et al., 2016*), and consequently attract significantly higher rates of pollinator visitation (*Blackmore & Goulson, 2014*). However, the plant species in wildflower seed mixtures vary greatly in their ability to provide nectar and pollen resources to foraging insects (*Hicks et al., 2016*), and this is reflected in differences in visitation rates by insects to these plots (*Ahrné, Bengtsson & Elmqvist, 2009*). Similarly, different wildflower mixes sown in agricultural margins support different communities of pollinators (*Williams et al., 2015*; *Warzecha et al., 2018*). Therefore, taxon specific knowledge of sown resource utilisation in urban areas will allow more specific recommendations for mixes to promote the abundance and diversity of each taxon of wild bees.

A number of studies have used DNA metabarcoding to study honeybee foraging choices by identifying pollen taken from honey samples (*De Vere et al., 2017*; *Hawkins et al., 2015*; *Richardson et al., 2015a*, *2015b*) and from pollen traps placed at the entrances to beehives (*Keller et al., 2015*). More recently, it has been shown that pollen samples taken directly from the bodies of pollinators can give an indication of foraging behaviour at the level of individual insects (*Bell et al., 2017*; *Lucas et al., 2018a*, *2018b*), although this requires killing the individuals sampled and thus may be problematic when sampling threatened species. The number of sequences obtained for a given plant species can offer a semi-quantitative picture of plant–pollinator interactions (*Pornon et al., 2016*). Sequencing-based identification of pollen is able to identify a greater number of taxa with better taxonomic resolution than morphological identification, and additionally reduces

the requirement for highly specialised taxonomic expertise (*Keller et al., 2015*; *Smart et al., 2017*).

Here, we investigate foraging preferences of bees feeding in sown wildflower strips using observational approaches coupled with metabarcoding of *rbcL*, a chloroplast gene. We aim to investigate (i) how the use of sown wildflower strips fits within use of the wider urban landscape, (ii) whether particular sown species are used preferentially over others, and (iii) whether the former two questions are affected by bee species identity. Additionally, the study aims to ascertain the effectiveness of non-destructive pollen sampling from individual bees as an alternative to killing sampled individuals when conducting pollen metabarcoding studies.

## METHODS

### Field sampling

Floral cover assessment, pollinator sampling and pollinator observations were carried out across 10 pollinator planting strip sites managed by Bournemouth Borough Council in July 2016 (Tables S1 and S2; Fig. S1). All data was collected between 9 am and 5 pm on dry days where the wind speed was less than 5 on the Beaufort scale and the temperature was above 15 °C.

At each site three 1 m$^2$ quadrats were placed by selecting the patches with the highest density of open flowers. The total floral cover as a percentage of each plant species was measured in these quadrats (vegetative growth was not recorded). At each of the three quadrats, a single 10 min pollinator count was carried out by recording all pollinators to enter the quadrat. Pollinators were identified by observation only, without netting. Honey bees and bumblebees were identified to species level, while other bees were identified to family level where possible and recorded as 'other solitary bees' if not.

On the same day as observational data was collected, pollinators were sampled whilst they were visiting flowers within wildflower strips. As abundances were not being measured, sampling continued until 15 individuals had been caught, regardless of how long this took. Pollen was non-destructively collected from pollinators by confining pollinators in sterile microcentrifuge tubes: 1.5 ml (for *A. mellifera* and *Bombus* spp.) or 0.2 ml (for Halictidae spp., hoverflies and beetles). Pollinators were contained in the tubes for 5 min each, in a cool place, allowing pollen to be deposited on the tube walls by insect movement, and then released. Pollinators were captured whilst feeding, and the species of flower visited was also recorded. *Bombus lucorum* and *B. terrestris* could not be distinguished in the non-destructive field during sampling, so they have been grouped under the name *B. terrestris* in this study. Similarly, individuals from the Halicidae could not be reliably identified in the field so individuals are classified at the family level.

### DNA extraction

DNA extraction was carried out following the method described by *Hawkins et al. (2015)*. Pollen was resuspended in 400 µl of buffer AP1 from a DNeasy Plant Mini Kit (Qiagen, Venlo, Netherlands), to which 80 µl of proteinase K (one mg ml$^{-1}$; Thermo Fisher Scientific, Waltham, MA, USA) was added alongside one µl of RNase A (100 mg ml$^{-1}$;

Qiagen, Venlo, Netherlands). Next, the samples were disrupted by shaking for four minutes at a speed of 30 1/s in a Retsch MM200 bead mill with custom adapter. Subsequent steps of the DNeasy Plant Mini Kit were carried out following manufacturer's instructions, with the omission of the QIAshredder column. Following extraction, DNA was stored at −20 °C.

To test whether the non-destructive method of sampling provided the same information as destructive sampling, 10 honeybees (*Apis mellifera*) were collected from outside hives in Dorchester (latitude: 50.719°, longitude: −2.419°) and six bumblebees (*Bombus terrestris*) individuals were collected on Bournemouth University campus (latitude: 50.741°, longitude: −1.894°). Collection was carried out on the 11th of November 2016 for honeybees, and 13th of February 2017 for bumblebees. Each of these individuals was both destructively and non-destructively sampled in a paired test: first, pollen was non-destructively collected from each individual as detailed under 'Field Sampling', and then the whole insect was transferred to a fresh tube and frozen at −20 °C prior to further processing. Pollen was 'washed' from each insect following the method employed by *Lucas et al. (2018a, 2018b)*: one ml of 1% sodium dodecyl sulphate (SDS) and 2% poly-vinyl pyrrolidinone (PVP) solution in water was added to the insect in each tube. Tubes were vigorously shaken by hand for 1 min, allowed to stand at room temperature for 5 min, and finally shaken vigorously by hand for a further 20 s. Next, the insect was removed and the tube containing pollen and the SDS-PVP solution was centrifuged at 13,000 rpm. Finally, the supernatant was removed and discarded, and DNA extraction was carried out as described above. Each extraction was tested by PCR amplification.

To prevent contamination, all DNA extractions were carried out in a laminar flow hood. Prior to each DNA extraction, surfaces within the hood were cleaned with 10% bleach followed by 95% ethanol, then all reagents and tools were placed within the hood and irradiated with UV light for at least 15 min. The hood was UV irradiated for 1 h every night. A negative control was included with each batch of extractions.

## Library preparation and sequencing

A section of the *rbcL* gene was amplified and prepared for sequencing following the protocol of *De Vere et al. (2017)*, adapted from that described by Illumina for the V4 region of 16S rRNA genes in bacteria (*Illumina, 2013*). The *rbcL* gene was chosen because a complete *rbcL* database has been created for native plants within Wales (*De Vere et al., 2012*), containing the majority of plants found in the UK as a whole. This protocol involves two PCR amplification steps: one to amplify the region of interest, and a second to add index and adapter sequences for sequencing. Following each PCR, samples are purified using AMPure beads. The final step involves library quantification, normalisation, and pooling. Only samples which produced a visible band in the first PCR step were carried through to further library preparation steps.

First, the *rbcL* gene was amplified using primers described in Table S3. Each reaction was at a final volume of 20 µl, and contained two µl of template DNA, Phusion High-Fidelity Master Mix at 1X concentration (New England Biolabs, Ipswich, MA, USA) and primers at 0.2 mM each. Thermal cycling conditions were as follows: 95 °C for 2 min;

35 cycles of 95 °C for 30 s, 50 °C for 1 min 30 s, 72 °C for 40 s (40 cycles); and 72 °C for 5 min. PCR clean-up was carried out using AMPure XP beads (Beckman Coulter, Brea, CA, USA) according to manufacturer's instructions. The second stage PCR was carried out at a final volume of 25 µl, with each reaction containing 12.5 µl Phusion High-Fidelity Master Mix, 2.5 µl Nexera XT Index Primer 1 (N7XX), 2.5 µl Nextera XT Index Primer 2 (S5XX), five µl water and 2.5 µl of purified product from the first PCR. Thermal cycling conditions were as follows: 95 °C for 3 min; eight cycles of 95 °C for 30 s, 55 °C for 30 s, and 72 °C for 30 s; and 72 °C for 5 min. A second PCR clean-up was carried out as described above. The product was quantified using a Qubit Fluorometer and Qubit dsDNA HS Assay Kit according to manufacturer's instructions, and pooled at equal concentrations to generate the final library pool. Prior to sequencing, the library was again quantified by Qubit and adjusted to 10 nM concentration with 0.1M Tris-HCl/0.01% Tween 20 solution, prior to denaturing and loading onto an Illumina MiSeq (Illumina, San Diego, CA, USA) following manufacturer's instructions.

Two negative controls were included in the sequence run: one containing a randomly chosen DNA extraction negative control, and one containing purified water.

### Data analysis

Sequencing data was analysed using a workflow previously described by *De Vere et al. (2017)* and available at https://github.com/colford/nbgw-plant-illumina-pipeline. Adapters and low-quality bases were trimmed using Trimmomatic (*Bolger, Lohse & Usadel, 2014*), then paired-end reads were merged using FLASH (*Magoc & Salzberg, 2011*). Singleton reads and merged sequences less than 450 bp in length were removed. Next, megablast (*McGinnis & Madden, 2004*) was used to search unique sequences against a custom BLAST database which consisted of all sequences from the Barcode Wales project (*De Vere et al., 2012*) alongside selected other sequences downloaded from GenBank (*Benson et al., 2012*). Results were manually filtered to remove plants that do not occur in the UK, based on *Stace (2010)*, and *Cubey & Merrick (2014)*.

All further analyses were carried out in R (*R Computing Team, 2017*). Rarefaction curves were generated using the R package 'vegan' (*Oksanen et al., 2017*), and rank-abundance curves in BiodiversityR. Bipartite pollinator–plant networks were drawn in R package 'bipartite' (*Dormann, Gruber & Fruend, 2008*). To test for significant differences in the number of reads and genus diversity between species and sites, generalized linear models were fitted with poisson (where no overdispersion was detected) or quasipoisson distributions using function 'glm'. Post hoc Tukey comparisons were carried out using package 'lsmeans' (*Lenth, 2016*).

## RESULTS

Of the 152 DNA extractions carried out on pollen taken from insects collected on urban pollinator strips, 41 produced a visible band after forty cycles of PCR and were sent for sequencing. A number of pollinators other than bees were collected, but none of these produced a band following PCR (Table 1). Of the insects collected for comparison of

**Table 1 Number of individuals of each taxa collected and sequenced per site.**

| Site number | Number Collected (Number sequenced in brackets) | | | | | | | | | | |
|---|---|---|---|---|---|---|---|---|---|---|---|
| | 18 | 21 | 22 | 23 | 24 | 28 | 29 | 30 | 31 | 33 | Total |
| Andrenidae | 2 (0) | 0 (0) | 0 (0) | 0 (0) | 0 (0) | 0 (0) | 0 (0) | 0 (0) | 0 (0) | 0 (0) | 2 (0) |
| *Apis mellifera* | 5 (1) | 4 (1) | 3 (1) | 7 (1) | 9 (2) | 3 (0) | 0 (0) | 4 (2) | 2 (1) | 2 (0) | 39 (9) |
| Coleoptera | 0 (0) | 0 (0) | 0 (0) | 0 (0) | 2 (0) | 0 (0) | 3 (0) | 0 (0) | 0 (0) | 3 (0) | 8 (0) |
| *Bombus hypnorum* | 1 (1) | 2 (0) | 0 (0) | 0 (0) | 0 (0) | 0 (0) | 0 (0) | 0 (0) | 0 (0) | 0 (0) | 3 (1) |
| *Bombus lapidarius* | 2 (0) | 2 (0) | 0 (0) | 2 (0) | 0 (0) | 0 (0) | 3 (0) | 4 (1) | 2 (0) | 2 (0) | 17 (1) |
| *Bombus pascorum* | 1 (1) | 1 (0) | 0 (0) | 1 (0) | 0 (0) | 0 (0) | 0 (0) | 0 (0) | 2 (1) | 0 (0) | 5 (2) |
| *Bombus terrestris* | 1 (0) | 6 (2) | 4 (4) | 4 (3) | 1 (0) | 4 (1) | 2 (2) | 6 (4) | 4 (3) | 4 (2) | 36 (21) |
| Diptera: Syrphidae | 1 (0) | 0 (0) | 2 (0) | 0 (0) | 0 (0) | 3 (0) | 0 (0) | 1 (0) | 2 (0) | 1 (0) | 10 (0) |
| Halictidae | 2 (1) | 0 (0) | 6 (2) | 1 (0) | 3 (1) | 5 (1) | 7 (1) | 0 (0) | 3 (0) | 5 (1) | 32 (7) |

**Note:**
The number collected is given first, followed by the number sequenced in brackets.

destructive and non-destructive sampling methods, five of six *B. terrestris* and one of 10 *A. mellifera* samples amplified successfully for both methods.

Sequencing yielded a total of 81,168,508 read pairs. Of these, 1,357,981 read pairs passed initial quality control and 98,985 were able to be paired. No reads from either negative control sample passed initial quality control, and no negative control produced a visible band following PCR amplification. Following manual filtering and removal of singleton reads, a mean of 1,131.0 (±178.0) reads per sample remained. Three samples yielded fewer than 100 reads and were excluded from all further analyses: these comprised two *A. mellifera* and one *B. terrestris* individual.

Rarefaction indicated that the number of reads required in order to detect the majority of genera varied greatly between samples (Fig. S2). However, rarefaction analysis indicated that sampling effort was sufficient to detect the majority of genera in most samples, despite variation in the number of reads. In most cases, a few plant genera made up the vast majority of reads with a longer 'tail' of genera that were only present at abundances of 5% or lower (Fig. S3). Based on this, a threshold of 5% of reads was considered to indicate 'major' food sources.

## Comparison of destructive and non-destructive sampling

To compare destructive and non-destructive sampling methodologies, DNA was collected using both methods from ten *A. mellifera* and six *B. terrestris* individuals. A single *A. mellifera* individual and four *B. terrestris* individuals produced visible bands following PCR amplification, and were thus sequenced. For destructive sampling, two *A. mellifera* and six *B. terrestris* individuals produced visible bands.

Overall, five genera were detected across all individuals collected for this part of the study: *Camellia*, *Erica*, *Hedera*, *Ulex*, and *Viburnum*. *Hedera* was the sole genus detected on *A. mellifera*, while *Camellia*, *Erica*, *Ulex*, and *Viburnum* were detected on *B. terrestris*. A minimum of 98% of genera in all pairs of destructively and non-destructively collected samples were shared between both samples (Table 2), and overall community composition was broadly similar between methods (Fig. 1). In all but one case, non-

**Table 2 Number of families detected using destructive and non-destructive sequencing, and the percentage of reads belonging to families that were detected using both sequencing methodologies.**

| Insect | Method | Total genera | >5% Reads | % Reads in genera detected by both methods |
|--------|--------|--------------|-----------|-------------------------------------------|
| AM | Non-destructive | 1 | 1 | 100 |
| | Destructive | 1 | 1 | 100 |
| BT1 | Non-destructive | 2 | 2 | 100 |
| | Destructive | 3 | 2 | 99 |
| BT2 | Non-destructive | 4 | 2 | 98 |
| | Destructive | 3 | 2 | 100 |
| BT3 | Non-destructive | 2 | 1 | 100 |
| | Destructive | 3 | 2 | 99 |
| BT4 | Non-destructive | 1 | 1 | 100 |
| | Destructive | 1 | 1 | 100 |
| BT5 | Non-destructive | 2 | 2 | 100 |
| | Destructive | 3 | 2 | 98 |
| BT6 | Non-destructive | 1 | 1 | 100 |
| | Destructive | 1 | 1 | 100 |

**Note:**
AM, *A. mellifera*; BT, *B. terrestris*.

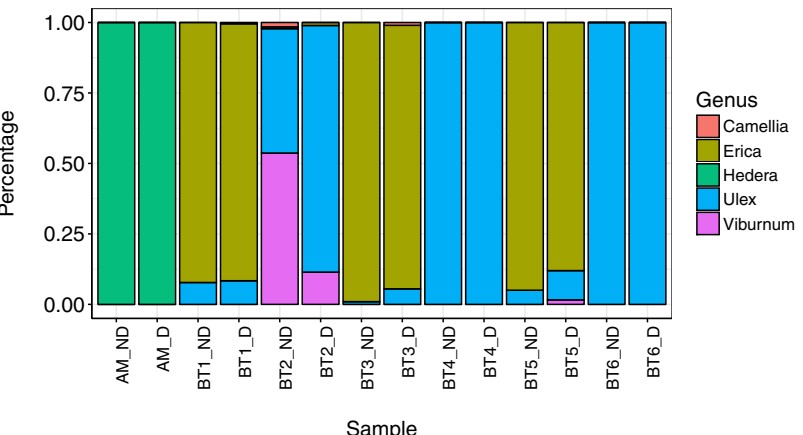

**Figure 1 Composition of pollen collected using destructive ('D') and non-destructive sampling methods ('ND').** The single *A. mellifera* individual that was successfully sampled using both approaches is named 'AM', while the six *B. terrestris* individuals are labelled BT1–BR6.

destructive and destructive sampling detected the same 'major' plant genera (defined as genera making up more than 5% of reads): the only exception to this was bee 'Bt3', where *Ulex* was detected in both samples but made up less than 1% of reads in the sample collected using non-destructive methods.

## Variation between bee species

The sequenced samples comprised 21 *B. terrestris* individuals, one individual each of *B. hypnorum* and *B. lapidarius*, two *B. pascuorum* individuals, nine *A. mellifera*

individuals, and seven belonging to family Halictidae. Both hoverflies (Diptera: Syrphidae) and beetles (Coleoptera) were collected, but did not yield useable DNA. The numbers of pollinators collected and sequenced at each site are described in Table 1.

The three most abundant pollinator taxa in the sequencing dataset (*A. mellifera*, *B. terrestris* and Halictidae spp.; Table 1) were chosen for intra-species and intra-site comparisons. Reads from *B. terrestris* belonged to the broadest range of plant genera, followed by Halictidae and then *A. mellifera* (Table 3). Of the plant genera which individual bees were collected from, metabarcoding detected five of seven plant genera visited by *B. terrestris*, two of the four genera visited by *A. mellifera*, and four of the seven genera visited by Halictidae spp. Additionally, metabarcoding detected a number of plant genera which were not present in wildflower plots (Table 3).

There were significant differences in genus richness of pollen found on individual insects of different species (resid. dev. = 36.1, d$f$ = 27, $p$ = 0.02) and at different sites (resid. dev. = 47.1, d$f$ = 34, $p$ = 0.03). In particular, pollen from *B. terrestris* individuals contained significantly more plant genera than that from *A. mellifera* ($z$ = 2.5, $p$ = 0.04). The mean number of plant genera per individual (±S.E.) was 4.2 (±0.4) for *B. terrestris*, 2.7 (±0.5) for *A. mellifera*, and 4.1 (±1.5) for Halictidae. The minimum number of genera detected on an individual was one for all bee species, with a maximum of seven for *B. terrestris*, five for *A. mellifera*, and 12 for Halictidae.

## Use of sown wildflower strips by pollinators

Metabarcoding of pollen collected from the bodies of bees detected pollen from a wide variety of plant families, many of which were not present in the wildflower plots (Fig. 2; Table 3).

In order to allow comparison between the observational and metabarcoding datasets, only data for bees which were sequenced is shown in Fig. 2A. Bees were collected from the flowers of nine different plant genera, all of which were flowering within wildflower strips at the time of sampling. The largest numbers of insects were collected on *Phacelia,* followed by *Centaurea* and *Chrysanthemum* (Fig. 2A). Conversely, while both *Phacelia* and *Chrysanthemum* were abundant in DNA metabarcoding data, *Papaver* made up a larger proportion of metabarcoding reads than of available floral resources (Fig. 2B) and *Centaurea* was not detected by DNA metabarcoding. In addition, a number of plant genera were detected on bee bodies, but were not detected in floral surveys: particularly abundant were *Ligustrum*, *Rosa*, and *Achillea* (Fig. 2B). These genera were often found on only a small subset of bees.

Plant genera present in wildflower plots accounted for the majority of reads (Fig. 3B), making up 69% of reads overall. Particularly well represented plant genera were *Papaver* and *Phacelia*. Conversely, the three members of the Asteraceae (*Anthemis, Centaurea,* and *Chrysanthemum*) found in sown wildflower plots were typically poorly represented amongst sequencing reads relative to the proportion of flowers available at each site: an exception to this was site 29, which was dominated by *Chrysanthemum* (Fig. 3) in the sequence reads. For three individuals, all pollen was assigned to a single genus that was

**Table 3 List of plant genera detected in sequenced pollen from each of the three most abundant pollinator species.**

| B. terrestris | | A. mellifera | | Halictidae | |
|---|---|---|---|---|---|
| Observed | Metabarcoding | Observed | Metabarcoding | Observed | Metabarcoding |
| | *Achillea* | *Achillea* | | | *Achillea* |
| | | | *Anthemis* | *Anthemis* | |
| *Borago* | | | | | |
| | | | | | **Brassica** |
| | *Buddleja* | | | | *Buddleja* |
| | **Campanula** | | | | |
| | *Centaurea* | *Centaurea* | *Centaurea* | *Centaurea* | *Centaurea* |
| | **Chelidonium** | | | | |
| *Chrysanth.* | *Chrysanth.* | *Chrysanth.* | | *Chrysanth.* | *Chrysanth.* |
| | **Cirsium** | | | | |
| | | | | | *Cosmos* |
| | | | | | **Crataegus** |
| *Echium* | *Echium* | | *Echium* | *Echium* | |
| *Escholzia* | *Escholzia* | *Escholzia* | | *Escholzia* | *Escholzia* |
| | | | | | **Fallopia/Polygonum** |
| | **Fuchsia** | | | | |
| | *Hydrangea* | | *Hydrangea* | | *Hydrangea* |
| | | | | | **Hypericum** |
| | | | *Lactuca* | | *Lactuca* |
| | | | | *Leucanthemum* | |
| | *Ligustrum* | | *Ligustrum* | | *Ligustrum* |
| *Linaria* | | | | | |
| | *Lupinus* | | *Lupinus* | | |
| | *Malva* | | *Malva* | | |
| | **Meconopsis** | | | | |
| | **Myosotis** | | | | |
| | | | | | **Oenothera** |
| *Papaver* | *Papaver* | | *Papaver* | *Papaver* | *Papaver* |
| | *Pentaglottis* | | *Pentaglottis* | | |
| *Phacelia* | *Phacelia* | | *Phacelia* | | *Phacelia* |
| | **Plantago** | | | | |
| | *Rosa* | | *Rosa* | | |
| | *Rubus* | | | | *Rubus* |
| | **Salvia** | | | | |
| | | | | | **Sambucus** |
| | **Symphytum** | | | | |
| | **Taraxacum** | | | | |
| | **Trachelium** | | | | |
| | *Trifolium* | | | | *Trifolium* |

**Notes:**

'Observed' interactions refers to all plant–pollinator interactions observed across three 10-min observation periods. Genera underlined were present in pollinator strips; bolded plant genera are unique to a single pollinator species. Chyrsanth., *Chyrsanthemum*.

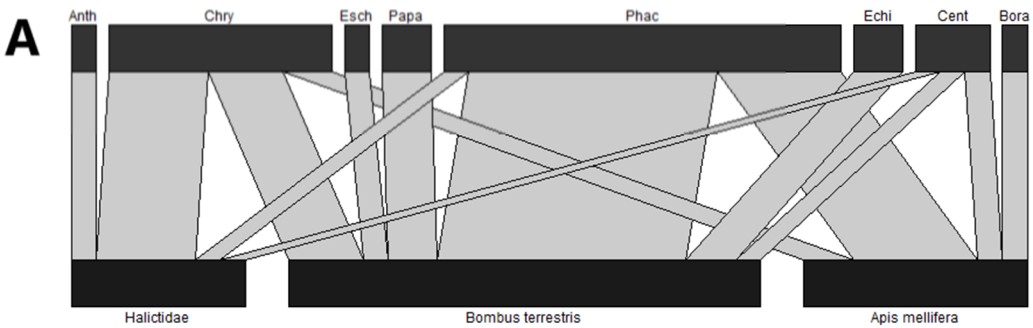

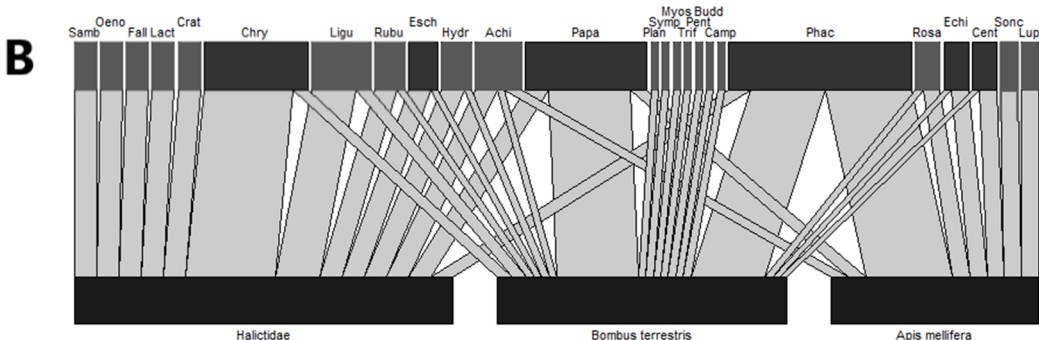

**Figure 2** **Bipartite network diagrams based on (A) observational data, that is, the number of bees captured on each plant genera and (B) the proportion of insects on which pollen from each plant genera was detected by metabarcoding.** Each plant genus was only counted as present on a given insect if it made up >5% of metabarcoding reads. In (A), only visitation data from bees whose pollen loads were sequenced is displayed. In (B), plant taxa which were present in the wildflower plots are coloured in dark grey, while other plant families are pale grey. Anth, *Anthemis*; Chry, *Chrysanthemum*; Esch, *Escholzia*; Papa, *Papaver*; Phac, *Phacelia*; Echi, *Echium*; Cent, *Centaurea*; Bora, *Borago*; Samb, *Sambucus*; Oeno, *Oenothera*; Fall, *Fallopia*; Lact, *Lactuca*; Crat, *Crataegus*; Ligu, *Ligustrum*; Rubu, *Rubus*; Hydr, *Hydrangea*; Achi, *Achillea*; Plan, *Plantago*; Symp, *Symphytum*; Myos, *Myosotis*; Camp, *Campanula*; Budd, *Buddleja*; Pent, *Pentaglottis*; Trif, *Trifolium*; Rosa, *Rosa*; Sonc, *Sonchus*; Lupi, *Lupinus*.

not present in the wildflower strips: instead, all sequences were assigned to *Achillea*, *Sonchus*, or *Rubus*. Other insects appeared to mix genera found within the wildflower plots with genera from external sources.

## DISCUSSION

In this study, we used high-throughput sequencing of the *rbcL* gene to characterise pollen collected from wild bees which were captured while foraging in sown wildflower strips within urban areas during July 2016, demonstrating the applicability of recently developed 'metabarcoding' techniques to assessing the effectiveness of conservation methods.

The majority of sequencing reads belonged to plant genera present in the wildflower strips, with particularly high abundances of *Phacelia* and *Papaver*. However, even though the individuals sampled were collected foraging in sown wildflower strips, bees were found to

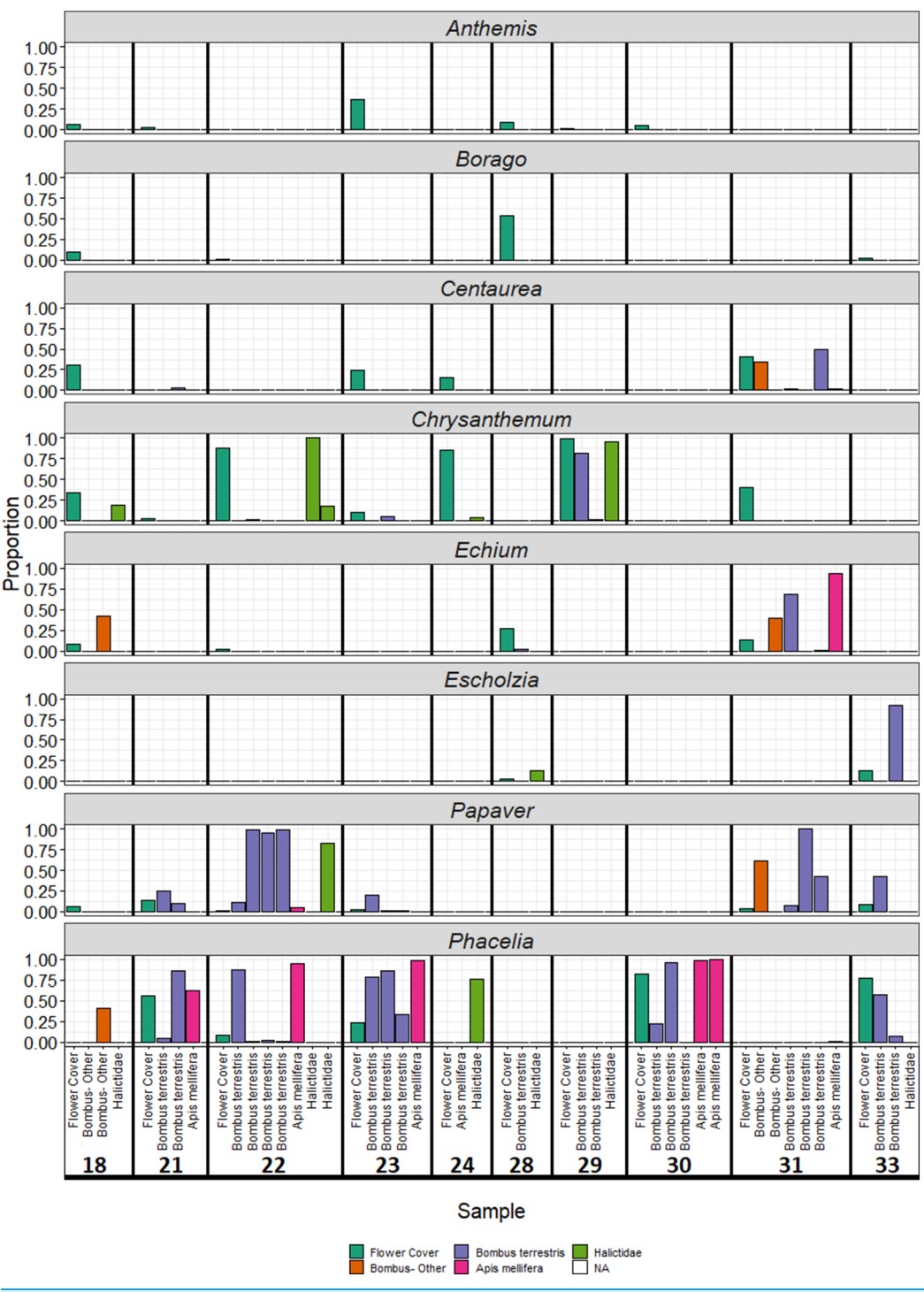

**Figure 3 Proportion of reads assigned to each plant species sown in wildflower strips by sample.** Colour of bars represents sample origin (floral survey or bee species).

utilise a wide range of plant families and genera, including some which were not present in the strips. This indicates that wildflower strips are only providing a proportion of the resources used by urban bee species.

## Comparison of destructive and non-destructive sampling

Previous studies have employed destructive sampling methods to collect pollen from individual insects (*Pornon et al., 2016*; *Lucas et al., 2018a*, *2018b*), in which bees are killed prior to pollen removal. While effective, this approach may be problematic under certain circumstances, for example, sampling of rare non-colony-forming pollinators, such as threatened species of hoverfly (*Ball & Morris, 2014*) or the sampling of bumblebee workers early in the season, which has lasting effects on colony reproductive success (*Muller & Schmid-Hempel, 1992*). The non-destructive sampling method employed in the current study involved trapping individual pollinators in Eppendorf tubes for long enough for pollen to fall off, then releasing the insect to continue foraging. However, using this approach the number of individuals yielding sufficient pollen for analysis was considerably lower, particularly for hoverflies and solitary bees, meaning that the utility of non-destructive sampling methods may be limited unless refinements can be made to improve insect contact with the collection tube.

Overall, where sufficient DNA was obtained for amplification with both methods, both destructive and non-destructive sampling of pollen gave broadly similar results. In all sample pairs, the vast majority of reads belonged to genera present in both samples. Additionally, in all but one case, the number of plant genera detected was the same in both destructively and non-destructively collected samples once rare genera were excluded. The proportion of reads assigned to each species was broadly similar between both methods, although some differences did exist: in particular, where multiple genera were present in a sample *Ulex* typically made up a greater proportion of reads when destructive sampling was used rather than non-destructive. It is possible that this is due to differences in the effectiveness of non-destructive pollen sampling in removing different pollen species, for example due to species-specific differences in pollen attachment (*Pacini & Hesse, 2005*). An additional factor to consider when comparing destructive and non-destructive sampling methodologies is that it is not possible to separate pollen stored in the scopa (pollen storage structure) from pollen found on the rest of the body. Pollen load analysis is commonly carried out on isolated scopal pollen in order to only include deliberately collected pollen rather than pollen accidentally collected when foraging for nectar (*Wood, Holland & Goulson, 2016*), or alternatively may be carried out with the scopa removed in order to only include pollen available for pollination (*Pornon et al., 2016*).

## Variation between bee species

Each bee species carried a unique range of plant genera (Table 3), similar to previous studies which have identified differences in floral preferences between pollinator species (*Geslin et al., 2013*; *Kells, Holland & Goulson, 2001*; *Leonhardt & Blüthgen, 2012*). The largest number of plant genera were detected on *B. terrestris* and the smallest number on Halictidae spp., although it should be noted that this pattern reflects the number of individuals sampled for each species. There were also differences in the choice of flowers used by each bee species: in observational data, Halictidae spp. were primarily observed feeding on Asteraceae such as *Chrysanthemum* and *Anthemis*, while *B. terrestris* and *A. mellifera* were observed to use *Phacelia*. Metabarcoding data detected similar species-

specific patterns of plant use: *Chrysanthemum* was detected on the highest proportion of Halictidae individuals, *Papaver* on *B. terrestris*, and *Phacelia* on *B. terrestris* and *A. mellifera*. This pattern reflects differences in tongue length: *B. terrestris* has a longer tongue than *A. mellifera*, which in turn has a longer tongue than Halictidae spp. (*Prys-Jones, 1982*; *Williams, 1997*). Halictidae individuals were found to feed primarily on open, brush and composite flowers, whose pollen and nectar is easier for short-tongued pollinators to access than tube-shaped flowers, for example, those of *Phacelia* (*Inouye, 1980*; *Willmer, 2011*). While there were very few observations and opportunities to sample mid- or long-tongued length bumblebees, the two *B. pascuorum* individuals (a longer-tongued species) which were caught with sufficient pollen loads for analysis were both found to be feeding predominantly on the tube-shaped flowers of the genera *Phacelia* and *Echium*.

The number of genera detected on individual bees by metabarcoding was unexpectedly high, particularly for *A. mellifera*. Earlier work suggests that *A. mellifera* foragers typically concentrate on a single plant species within a given foraging trip, while *Bombus* spp. and Halictidae feed on multiple species per foraging trip (*Beil, Horn & Schwabe, 2008*; reviewed by *Grant, 1950*; *Leonhardt & Blüthgen, 2012*; *Wood, Holland & Goulson, 2016*; but c.f. *Brodschneider et al., 2018*). Conversely, in the current study *A. mellifera* individuals carried pollen from an average of 2.7 plant genera. However, in many samples diversity was increased by plant genera which made up only a small proportion of reads: while metabarcoding data is only semi-quantitative (*Elbrecht & Leese, 2015*; *Richardson et al., 2015a*), the low abundance of reads from these genera raises the possibility that they do not represent genuine food plants, and are instead a result of pollen grains left over from previous foraging trips or transferred from the bodies of other bees in the hive or nest (*Free & Williams, 1972*). Alternatively, these plants may represent secondary, or 'minor', food sources, which may become more important as some pollinators exhibit more generalist foraging strategies in urban habitats (*Geslin et al., 2013*).

## Use of sown wildflower strips by pollinators

All species of bee included in the study were found to utilise a range of plant genera which were not present in pollinator strips, demonstrating the ability of pollen identification to detect plant–pollinator interactions on a broader spatial scale then could be easily achieved with observation alone, especially given that honeybees, solitary bees and bumblebees can all forage over long distances (*Beekman & Ratnieks, 2000*; *Knight et al., 2005*; *Zurbuchen et al., 2010*). Overall, five of the 21 *B. terrestris* pollen loads were dominated by non-sown genera (>50% of reads): this is comparable to *B. terrestris* individuals foraging in sown wildflower strips on arable land (*Carvell et al., 2006*). Outside of wildflower strips, the majority of plant genera detected contained native members: for example, *Rubus* and *Rosa*. Previous work has found that native species are often preferred by pollinators (*Corbet et al., 2001*; *Pardee & Philpott, 2014*; *Salisbury et al., 2015*). Most of the plant genera originating from outside wildflower strips are commonly found in parks and gardens, either cultivated (e.g. *Lupinus*, *Hydrangea*, *Buddleja*, *Ligustrum*) or as wild plants (e.g. *Rubus*, *Sonchus*, *Lactuca*). Gardens provide a large proportion of urban

green space (*Loram et al., 2007*; *Thompson et al., 2003*) and contain diverse plant species (*Gaston et al., 2005*). Alongside previous studies (*Matteson & Langellotto, 2009*; *Osborne et al., 2008*), the presence of garden species in the current study highlights the importance of gardens for urban bees. The abundance of species from outside the wildflower strips suggests that while wildflower strips alone are not enough to provision nearby bees adequately, bees are able to flexibly utilise a wide range of plant taxa and urban landscape features in order to obtain adequate floral resources.

Despite the fact that all the insects studied here were collected in urban pollinator strips, there was a relatively weak correspondence between the floral composition of the strips (i.e. which plants were flowering in strips at the time of sampling) and the composition of pollen collected from bees. The plant genera detected in the highest proportion of pollen samples were *Chrysanthemum*, *Papaver*, and *Phacelia*, suggesting that these plants may be valuable contributors to wildflower mixes sown to support urban bees. Each of these genera established well and produced large quantities of flowers at the time of sampling. *Phacelia* in particular is a common component of wildflower mixes sown in agricultural margins, and is often a significant component of foraging on these margins, particularly for *B. terrestris* (*Carreck & Williams, 1997*; *Kells, Holland & Goulson, 2001*; *Pywell et al., 2005*). Members of family Asteraceae are the plants most visited by small bees in agricultural margins (*Wood, Holland & Goulson, 2017*). In the current study, similar patterns were found: *A. mellifera* and *B. terrestris* disproportionately carried pollen from *Phacelia* and *Papaver*, while Halictidae spp. carried pollen from *Chrysanthemum*. Therefore, a mixture of these species appears to provide floral resources for the range of Hymenoptera studied. *Phacelia* and *Papaver* both contain high levels of protein and essential amino acids (*Hanley et al., 2008*; *Roulston, Cane & Buchmann, 2000*; *Weiner et al., 2010*), although this is not the case for *Chrysanthemum* (*Roulston, Cane & Buchmann, 2000*). However, it should be noted that all sampling was carried out in July, and it is likely that the relative contribution of different plant species to bee foraging varies over the course of the season.

In several cases, bees were observed to feed on plant genera within the wildflower strips that were not present in metabarcoding data: examples include *Borago*, *Linaria*, *Leucanthenum*, and *Anthemis*. Only DNA extractions which produced a band following PCR amplification were processed for sequencing, and this may have biased sequencing towards samples which were taken from pollen-foraging individuals since these individuals carry more pollen grains and thus are more likely to yield adequate DNA for amplification. In *A. mellifera*, individual bees are specialised for either pollen or nectar collection (*Robinson & Page, 1989*). Similarly, while Halictidae and *B. terrestris* females do not show individual specialisation, they may forage exclusively for either nectar or pollen on separate trips (*Batra, 1964*; *Delph & Lively, 1992*; *Konzmann & Lunau, 2014*). In the current study, pollen stored in the scopa was not separated from pollen on the rest of the body, and so it is not possible to distinguish deliberately collected pollen from pollen accidentally collected while foraging for nectar. However, previous studies show that pollen-feeding insects may make different foraging choices to nectar-foraging individuals: in particular, Asteraceae and Boraginaceae (families containing *Anthemis*,

*Leucathernum*, and *Borago*, which were represented in observational but not metabarcoding data) are heavily used for nectar but not for pollen by bumblebees (*Goulson et al., 2005*), although it should be noted that small quantities of each family are found in bumblebee pollen loads (*Kleijn & Raemakers, 2008*).

However, detection may have been inhibited by methodology: while family Boraginaceae is well-detected by the primers chosen (*De Vere et al., 2012*), the amplification efficiency of different species in mixed samples is variable (*Pornon et al., 2016*). Additionally, *rbcL* may offer only poor discrimination for Asteraceae due to low levels of interspecific divergence (*Gao et al., 2010*), which were highly represented in wildflower plots. While at least one member of family Asteraceae (*Chrysanthemum*) was confidently identified to genus level in sequence datasets, a large number of reads could only be assigned to Asteraceae at the family level and may originate from species that were observed in floral surveys but not metabarcoding data (e.g. *Anthemis*).

## CONCLUSIONS

Despite the fact that all bees sampled were collected in wildflower strips, a number of them were found to utilise species not present in wildflower strips, highlighting the role that gardens play in providing adequate floral resources for urban bees. Within wildflower strips, both DNA metabarcoding data and observational data suggested that *Phacelia* and *Chrysanthemum* were particularly important genera for bees at the time of sampling, while metabarcoding additionally suggested that *Papaver* was also an important source of pollen for insects. Different bee species used different plant genera, highlighting the importance of including a range of plants in foraging strips: at the time of sampling, *Papaver* was used by the highest proportion of *B. terrestris* individuals, *Phacelia* by both *A. mellifera* and *B. terrestris*, and *Chrysanthemum* by Halictidae spp. However, all samples were collected in July and it is likely that other plants become more important during other times of the year. Finally, we show that non-destructive sampling coupled with DNA metabarcoding can be used to evaluate the ways in which pollinators interact with sown wildflower strips in urban environments, although it produces fewer successful samples compared to destructive methods.

## ACKNOWLEDGEMENTS

Thanks to Robert Potter and Mark Holloway of Bournemouth Borough Council for provision of wildflower strips, details of seed mixes and permission to collect samples. Thanks to Arne Loth and Kimberley Tickner for assistance in liaising with the council and locating and mapping the sown wildflower areas.

### Funding

This work was supported by a Bournemouth University Higher Education Innovation Fund Grant to Dr Elizabeth Franklin. The funders had no role in study design, data collection and analysis, decision to publish, or preparation of the manuscript.

## Grant Disclosures

The following grant information was disclosed by the authors:
Bournemouth University Higher Education Innovation.

## Competing Interests

Natasha de Vere is an Academic Editor for PeerJ.

## Author Contributions

- Caitlin Potter performed the experiments, analysed the data, prepared figures and/or tables, authored or reviewed drafts of the paper, approved the final draft.
- Natasha de Vere conceived and designed the experiments, analysed the data, contributed reagents/materials/analysis tools, authored or reviewed drafts of the paper, approved the final draft.
- Laura E. Jones performed the experiments, analysed the data, approved the final draft.
- Col R. Ford analysed the data, contributed reagents/materials/analysis tools, approved the final draft.
- Matthew J. Hegarty conceived and designed the experiments, performed the experiments, contributed reagents/materials/analysis tools, approved the final draft.
- Kathy H. Hodder conceived and designed the experiments, performed the experiments, approved the final draft.
- Anita Diaz conceived and designed the experiments, performed the experiments, approved the final draft.
- Elizabeth L. Franklin conceived and designed the experiments, performed the experiments, authored or reviewed drafts of the paper, approved the final draft.

## Data Availability

Sequencing data was analysed using a workflow available at https://github.com/colford/nbgw-plant-illumina-pipeline.

Raw sequence data is available on the Sequence Read Archive (SRA) at PRJNA481887

Potter, Caitlin (2019): Raw fastq files. figshare. Fileset. https://doi.org/10.6084/m9.figshare.6893501.v1.

Potter, Caitlin (2019): Metadata. figshare. Dataset. https://doi.org/10.6084/m9.figshare.6930857.v1.

## Supplemental Information

Supplemental information for this article can be found online at http://dx.doi.org/10.7717/peerj.5999#supplemental-information.

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
