# Peer review of "Pollen metabarcoding reveals broad and species-specific resource use by urban bees"

_PeerJ, doi:10.7717/peerj.5999_

## Round 0.1 · original submission · Major Revisions

Comments from the Editor: I suggest that you strongly consider reviewer advice on stronger justification and argument balance and progression through the introduction.

Also, please provide more detail for some of your sampling techniques, especially for the statistical techniques later presented or claimed (e.g. difference in abundance). Pay close attention to the request for assurance on pseudo-replication as samples should have been pooled?
Most importantly, there is a critical need to discuss your results in light of not sampling specifically from the scopa – which is where many bears store the pollen they purposefully collect. This must be opened up and discussed in light of the non-destructive sampling technique used.
Consider replacing pollinator with bee as this study only presents meaningful results for that taxa of pollinators.

Also, please consider title changes to ‘resource’ rather than 'habitat' – as habitat is far more than just nectar/pollen resource; and consider combining tables 3 and 4. I look forward to receiving your revised manuscript.

Reviewer 1 ·

Basic reporting

The language used if fairly clear throughout, though there are a few spelling mistakes that need to be checked.The introduction is reasonable but could use a little more nuance when setting the stage for global bee declines and the role that urban management can play in this overall picture. The layout and structure conforms to PeerJ standards (though see acknowledgements). The results presented are relevant to the hypotheses.


Title – ‘habitat’ use. Since you present data on use of different pollens by bees, the use of the word habitat is slightly perplexing to me. Surely ‘resource’ use would be more appropriate?

Introduction

L45-48. Use of ‘severe’ is quite strong. Most of the data for wild bee declines comes from bumble bees, and indeed you cite four bumble bee papers in this sentence, all focusing on North America and Europe. I would take more time to specify that you are talking about wild bees, since you later go on to present data on honey bees, and find additional citations to more broadly support wild bee and ‘other pollinator’ declines.

L57. The Baldock paper found higher species richness of wild bees in urban areas compared to rural areas, so this is a surprising citation. Overall, the principle that urban areas have lower species richness than rural areas is contested, with papers on both side of the argument.

L53-55. The topic of urbanisation comes out of the blue here. You have not introduced urbanisation as a factor that can or could cause bee decline. Perhaps a citation to explain that urbanisation is expanding – and is it expanding in the western world, the only area which your papers demonstrate bee declines have occurred?

L59-60. Check formatting.

L66. Significantly

L71. Carreck and Williams 1997 is a little out of date perhaps

L73. Most of the differences in community visitation in these studies stems from the constituent wildflowers chosen. Many bee species have narrow ecological niches, and hence will respond differently to different seed mixes as they may or may not provide their preferred forage plants. When you write that “will help to ensure that utility for pollinators is maximised” this is highly dependent on what your desired outcome is – e.g. helping bumble bees or solitary bees for example. I would make your wording a little more precise – do you want to build a species specific picture of sown resource utilisation in urban areas, since this seems to be the knowledge gap as currently written? If so, specify this.

L89-91. This destructive/non-destructive concept would benefit from some introduction

Acknowledgements should not be used to report funding, that information will appear in a separate Funding Statement on the published paper, see guidelines.

Experimental design

I do not have sufficient experience with the metabarcoding technique to assess this part of the paper.

Methods

L99 total cover or relative cover?

L100-101. Were bees observed only or also netted? If observations, were bees placed into groups, e.g. small bee, bumble bee, honey bee, and so on? Also, it is not quite clear, did you have 3 10-minute counts at each quadrat (total 90 minutes) or a single 10-minute count at each quadrat (total 30 minutes)?

L102. How long was this collection period?

L117. 30 1/s, is this a rate, and how long were they shaken for?

L125. Check date format against journal standards

Validity of the findings

Results

L231. I’m not sure why you report an R2 since a permanova is only saying whether the communities are different. How can you have an explanatory factor?
There was no mention of PERMANOVA or NMDS analysis in the methods. When looking at the NMDS plot, you have three dots for each site – I assume this is one for each quadrat? This is not in the methods, and strictly speaking samples should be pooled per site (since data was collected in the same place on the same day) otherwise you have pseudoreplication (subsampling from the same population) and hence you can’t actually do NMDS community analysis since you don’t have a grouping factor (n=1 per site) such as seed mix a, seed mix b and so on.

L232-235. Counts during the pollinator surveys? What statistical test did you use to test for observed difference in abundance? You do not specify it here or in the methods – just referring to reads and genus diversity which I assume refers to the sequencing dataset.
L249. Minimum of one – for a single individual?

L252-254. You have to be very careful here. Up to this point you have been referring to pollen removed from the bodies of pollinators. You now state that pollinators collected pollen from. This is not something you can assume. Many, many, many bee species (if not all) visit more flower species for nectar than for pollen. Some of the pollen types you detect with the very sensitive metabarcoding approach will be from nectar plants only. Since you did not use destructive sampling from the bees collected from the wildflower plots, you cannot be certain if the bee was collecting pollen or it was simply on its body from a previous nectar visit (as opposed to removing pollen directly from the scopa to ensure that it was purposefully collected). You will be able to infer a certain amount from the number of reads but you must be careful to note that the technique as you use it has clear flaws when trying to unambiguously identify pollen sources as opposed to plants used as nectar sources only.

L262-263. Do you mean detected by metabarcoding but not detected by floral surveys?

Discussion

L308. Table 4 doesn’t seem particularly relevant to this particular point

L327-340. It is important to discuss a third reason – pollen grains present on the body from nectar visits. Honey bees are indeed predominantly flower constant when pollen gathering, so multiple detections are highly suggestive of pollen accumulated on the body from nectar visits.

L392. This is a minor point but Asteraceae is used by bumble bees for pollen in small quantities, see Kleijn and Raemakers 2008 Ecology. I also feel it is worth discussing the difficulty in distinguishing between the very similar Anthemis and Chrysanthemum using the barcoding technique a little more.

Acknowledgements should not be used to report funding, that information will appear in a separate Funding Statement on the published paper, see guidelines.

Figures and Tables
Standardise formatting of the tables.
Table 1. This should be moved to supplementary information
Table 3. Requires additional formatting. “Site number” should be added as an additional row above the numerical sequence
Table 4. Check spelling of Andrena. Why is ‘male’ specific for certain taxa? If sex is specified, an additional column should be added. The table heading is ambiguous and cannot be interpreted without reference to the text. Sequenced how – to determine the insect species identity, or the identity of pollen on their body? “proportion of pollinators that produced a visible band and were successfully sequenced” is not correct, since you present a total not a proportion. Moreover there is a difference between the number successfully sequenced (heading) and the number sequences (table).
Table 5. Check spelling of Leucanthemum

Additional comments

I enjoyed reading this paper as it provides additional resolution to the flowering resources used by some species of urban bees visiting patches of annual flowers in urban areas. I think there are valuable results presented here, specifically in highlighting the value of annual flowers for bees (namely poppy and small flowered Asteraceae) and that these bees also use a wider variety of plant species found across the urban environment. As outlined in my comments above, I feel that the introduction needs a bit of work to more smoothly introduce the concepts and hypotheses you deal with.

Reviewer 2 ·

Basic reporting

The manuscript is well-written with appropriate background. Raw data is shared on NCBI SRA.

Experimental design

The tables could be reorganized to make them more easily understood

Table 1: Include summary of seed mix components in table heading?
Would be more broadly accessible to provide latitude/longitude rather than grid reference.
Table 2: The meaning of the column “% Reads in shared genera” is not clear. Please change or provide additional information in the table to help interpret
Tables 3 and 4 should be combined into a single table so that the reader can compare number of bees collected with number yielding rbcL PCR products.

Validity of the findings

The data reported in the manuscript was collected only in July -- it should be made clear in the abstract and throughout the manuscript that these results are from a relatively limited period of time and that there could be differences outside of this time period.

Additional comments

I question the use of the term “pollinator” throughout the manuscript when “bee” would be more descriptive of the taxa sampled. “Bee” already takes the place of “pollinator” in the title. I understand that the intention of the research was to include flies and other insect groups, but the results presented are almost exclusively about bees.

Abstract: Include mention of rbcL in abstract so readers immediately know which gene was used
Introduction:
Line 86: mention that rbcL is a chloroplast gene
Methods:
Line 105: provide a bit more detail about sample tubes -- were these standard plastic microcentrifuge tubes?
Line 109: Rephrase “not be distinguished in the non-destructive field during sampling”
Line 121: Remove “that”
Line 212: Should be “six _Bombus terrestris_ individuals”?
Line 223-224: This was already reported in lines 189-190
Discussion:
Lines 383-385: Please reword to be more explicit. This would have biased analyses against nectar-foraging bees because there would not have been sufficient pollen to produce a band on the gel?

---

## Round 0.2 · accepted · Accept

Both reviewers were very happy with the responses and improvements that you have made to your manuscript, so I am more than happy to provide an Accept decision. There are several minor comments that both reviewers picked up, please attend to these during the production phase or in response to PeerJ staff communication.

# Reviewer 1 ·

Basic reporting

no comment

Experimental design

The introduction has been rewritten to more thoroughly introduce the question that the work addressed and to set it within a wider context.

Validity of the findings

The statistical methods employed have been improved from the original submission.

Additional comments

Thank you for your work addressing the concerns from the first review. I am very happy to see this work published in its current form, though there are a couple of very tiny changes that are needed but I don't think that merits the minor revisions category.

L37-40. Repetition of certain, consider altering
L42. Check paragraph formatting
L47-48. First example is within brackets with the references whilst the following examples are outside brackets
L112. Double full stop
L118. Check sentence end for missing full stop
L235. Change 7 to seven and 9 to nine
L266-267. Specific numbers?
L385. Remove c.f.

Reviewer 2 ·

Basic reporting

I previously reviewed this manuscript and I find that the authors did a nice job improving the manuscript and responding to both my comments and the comments of the other reviewer. Overall, I think this is a really nice paper and demonstrates the kind of ecological and practical applications for which pollen metabarcoding is very suitable. The discussion of the weaknesses of the non-destructive collection methods are nuanced and include needed discussion of incidental pollen (though that is still good evidence that a plant is providing benefits to bees through provision of nectar).

Take a careful look at all Table/Figure references. I caught a few that appeared to be incorrect.

L224: Should this be Table 1 referenced here?
L265: Again, I think this is referring to Table 1.

Experimental design

As before, the experimental design appears to be good and the experiments thoughtfully executed.

The “Data Accessibility” section has been removed in the revised version. It is important that the reference to the NCBI submission remains in some form in the manuscript, whether here or in the Methods section.

L150-153: Report lat/lon to the thousandths place as in Table S1.

Validity of the findings

As before, the findings appear to be valid and the changes made have improved the manuscript.